

# Preoperative fibrinogen level predicts the risk and prognosis of patients with native valve infective endocarditis undergoing valve surgery

Jia Li[1,*], Junyong Zhao[2,*], Ning Sun[2], Lijiao Zhang[1], Qing Su[1], Wei Xu[1], Xiaolin Luo[2], Zhichun Gao[2], Keting Zhu[1], Renjie Zhou[1] and Zhexue Qin[2]

[1] Department of Emergency, Xinqiao Hospital, Army Medical University, Chongqing, China
[2] Department of Cardiology, Xinqiao Hospital, Army Medical University, Chongqing, China
* These authors contributed equally to this work.

## ABSTRACT

**Aim:** The aim of this study was to assess the clinical significance and prognostic value of the preoperative fibrinogen (FBG) level in patients with native valve infective endocarditis (NVIE) who underwent valve surgery.

**Methods:** This retrospective study included a total of 163 consecutive patients who were diagnosed with NVIE and underwent valve surgery from January 2019 to January 2022 in our hospital. The primary endpoint was all-cause mortality.

**Results:** All-cause mortality was observed in 9.2% of the patients ($n = 15$). Body mass index (BMI) was lower in the survival group ($p = 0.025$), whereas FBG ($p = 0.008$) and platelet count ($p = 0.044$) were significantly greater in the survival group than in the death group. Multivariate Cox proportional hazards analysis revealed that FBG (HR, 0.55; 95% CI, [0.32–0.94]; $p = 0.029$) was an independent prognostic factor for all-cause mortality. Furthermore, Kaplan–Meier survival curve analysis revealed that patients with low FBG levels (<3.28 g/L) had a significantly greater mortality rate ($p = 0.034$) than did those with high FBG levels (>3.99 g/L). In the trend analysis, the FBG tertiles were significantly related to all-cause mortality in all three adjusted models, and the $p$ values for trend were 0.017, 0.016, and 0.028, respectively.

**Conclusion:** Preoperative FBG may serve as a prognostic factor for all-cause mortality, and an FBG concentration less than 3.28 g/L was associated with a greater risk of all-cause mortality in NVIE patients undergoing valve surgery.

Corresponding authors
Renjie Zhou, zhou_rj@aliyun.com
Zhexue Qin, zhexueqin@126.com

## INTRODUCTION

Infective endocarditis (IE) is a relatively rare infectious disease with a crude incidence ranging from three to seven cases per 100,000 person-years according to contemporary population surveys (*Baddour et al., 2015*; *Cahill & Prendergast, 2016*). IE is still a potentially life-threatening disease whose outcome has not significantly improved despite novel diagnostic and therapeutic strategies (*Habib et al., 2015*; *Holland et al., 2016*).

Owing to the increasing number of IE patients undergoing surgery, a greater proportion of patients with prosthetic valves and other cardiac devices have developed endocarditis (*Lalani et al., 2010*). However, native valve infective endocarditis (NVIE) is still the most common type (*Chambers & Bayer, 2020*). Surgical therapy is currently performed in nearly 50% to 60% of NVIE patients. The specific indications for early valve surgery are progressive valve and tissue damage, uncontrolled infection and a high risk of embolism. The indications for surgery have been predominantly derived from historical observational studies (*Wang, Gaca & Chu, 2018*; *Cahill et al., 2017*). However, how to identify patients who are most likely to benefit from early valve surgery remains unclear. Thus, the identification of novel prognostic biomarkers is needed to identify NVIE patients who might benefit from more aggressive surgical therapy.

The mechanisms of NVIE pathogenesis are still insufficiently understood, but the immune response, bacterial infection and coagulation system of the human body have been shown to be involved (*Werdan et al., 2014*). An endocarditis lesion is formed mainly by fibrin and platelet blood clots infested with bacteria that adhere to the cardiac valves (*Liesenborghs et al., 2020*). Fibrinogen and fibrin, the key components of blood clots, play crucial roles in coagulation and thrombosis (*Kattula, Byrnes & Wolberg, 2017*). The coagulation system is significantly involved in IE by mediating the initial adhesion of bacteria to cardiac valves, facilitating the development and maturation of vegetation, and inducing complications such as embolization and valve destruction (*Durante-Mangoni, Molaro & Iossa, 2014*).

Fibrinogen levels have been found to be predictive of cardiovascular diseases because they are strongly associated with the risk of thrombosis and bleeding (*Fibrinogen Studies et al., 2005*; *Koenig, 2003*). One observational study revealed a significant decrease in fibrinogen and thrombodynamics in patients with fatal IE (*Koltsova et al., 2021*). Another study revealed that fibrin formation is impaired during cardiac surgery, which suggests that fibrinogen deficiency may be an important cause of ongoing bleeding in both adult and pediatric cardiac surgery patients (*Solomon, Rahe-Meyer & Sorensen, 2011*). These findings suggest a potential connection between the plasma fibrinogen level and the progression of NVIE. We conducted a retrospective study to evaluate the clinical significance and prognostic value of fibrinogen in patients with NVIE undergoing surgery. Our results may help surgeons assess the risk and prognosis of early valve surgery in patients with NVIE.

## MATERIALS AND METHODS

### Study population

In this observational and retrospective study, patients who were diagnosed with IE by two independent cardiologists on the basis of the modified Duke criteria from January 2019 to January 2022 were consecutively enrolled at Xinqiao Hospital, Chongqing, China. The inclusion criteria for the cohort were as follows: (1) had a diagnosis of definite IE (or possible IE considered and treated as IE), (2) were aged ≥18 years, and (3) had undergone valve surgery in the hospital. Patients with cardiac device-related IE, prosthetic IE, or a lack

of medical records and those who received antibiotic therapy alone were excluded. The study was approved by the Medical Ethics Committee of the Second Affiliated Hospital of Army Medical University (No. 2023-Research 109-01). All procedures were carried out in compliance with the Declaration of Helsinki and its amendments. In view of the observational and retrospective study design, no written informed consent was necessary.

## Data collection

Data on demographics, clinical presentations, hospitalizations, symptoms, comorbidities and medical procedures were extracted by reviewing medical and nursing records. All hematologic and biochemical parameters were measured in venous blood samples drawn before surgery. Measurements of coagulation parameter levels were performed by using a fully automated coagulation analyzer (ACL TOP 700 LAS; Instrumentation Laboratory Co., Bedford, MA, USA). FBG levels were measured *via* the HemosIL Fibrinogen-C XL (Instrumentation Laboratory Co., Bedford, MA, USA).

## Follow-up and endpoints

The primary endpoint was all-cause mortality. Data on survival status and adverse events were obtained with a detailed review of all available electronic medical records, telephone interviews or office visits with the patients or their family members. Loss to follow-up was defined as failure to confirm the status of patients due to insufficient medical records and a lack of response to telephone contact.

## Statistical analysis

Continuous data are expressed as the means ± SDs or medians and interquartile ranges (IQRs). The categorical variables are expressed as percentages and counts. Comparisons between groups were performed *via* two-sample t tests or Mann–Whitney U tests for continuous data and $\chi^2$ tests or Fisher's exact tests for categorical data. Survival analysis was performed *via* Kaplan–Meier curves, which were compared *via* the log-rank test. Multivariate Cox proportional hazards regression analysis was used to identify potential prognostic factors by including variables with $p < 0.05$ in the univariate analysis. Multivariate analysis was performed with the backward conditional method.

Trend analysis was performed by using stratified Cox regression models to calculate HRs and 95% CIs for all-cause mortality by entering the median value for each category of the FBG tertiles. The Cox regression models were adjusted for age, sex, body mass index (Model 1), platelet count, left ventricular ejection fraction (Model 2), activated partial thromboplastin time, thrombin time and the prothrombin time international normalized ratio (Model 3). The predictive performance of FBG for all-cause mortality in patients with NVIE was evaluated by the area under the receiver operating characteristic (ROC) curve (AUC), and the cutoff values were determined. $p$ values < 0.05 were considered to indicate statistical significance. Statistical analysis was performed *via* the statistical software IBM SPSS version 26.0 (SPSS, Inc., Chicago, IL, USA).
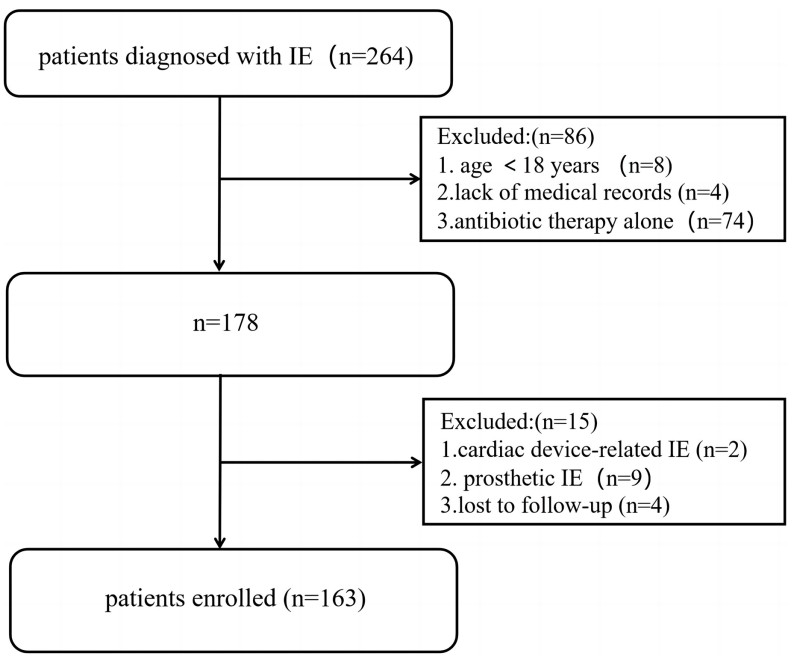

**Figure 1 Study flow chart of patients enrollment.**

## RESULTS

### Patient characteristics

We collected the medical records of 264 consecutive patients with a diagnosis of IE, 163 of whom were included in the study according to the inclusion and exclusion criteria (Fig. 1). The median follow-up time was 1,062 (IQR: 802–1,400) days. None of the patients underwent anticoagulant treatment on admission. The mean age of the cohort was 46.6 ± 12.4 years, and 74.2% ($n = 121$) were male. All-cause mortality was observed in 9.2% of the patients ($n = 15$), and the patients were divided into two groups according to survival status. The baseline characteristics of the patients before surgery are presented in Table 1.

For valve involvement, aortic valve endocarditis was most frequently observed (54.0%, $n = 88$), followed by mitral valve (29.4%, $n = 48$) and both aortic and mitral valve involvement (9.8%, $n = 16$). Analysis of blood cultures at admission revealed positive results in 28.2% of the patients ($n = 46$), and the most frequent microorganism detected was *Streptococcus* ($n = 31$). Body mass index (BMI) was significantly lower in the survival group ($p = 0.025$), whereas FBG ($p = 0.008$) and the platelet count ($p = 0.044$) were significantly greater (Table 1).

### Coagulation parameters and all-cause mortality

Coagulation parameters were evaluated in relation to all-cause mortality in IE patients *via* the Cox proportional hazards model (Table 2). Univariate analysis revealed that the PT-INR ($p = 0.001$), FBG level ($p = 0.009$) and PLT ($p = 0.033$) were significantly correlated with all-cause mortality. Moreover, PT-INR (HR, 8.94; 95% CI, [1.83–43.75]; $p = 0.007$) and FBG (HR, 0.55; 95% CI, [0.32–0.94]; $p = 0.030$) were identified as

**Table 1 Baseline characteristics of the NVIE patients before surgery.**

| | Total N = 163 | Death group N = 15 | Survival group N = 148 | p value |
|---|---|---|---|---|
| **Patient characteristics** | | | | |
| Age, years | 46.6 ± 12.4 | 48.8 ± 13.0 | 46.3 ± 12.3 | 0.480 |
| Gender (male) | 121 (74.2%) | 11 (73.3%) | 110 (74.3%) | 1.000 |
| BMI, kg/m$^2$ | 22.1 ± 3.2 | 23.9 ± 2.7 | 21.9 ± 3.2 | 0.025* |
| Hospital stay (days) | 22.2 ± 8.5 | 20.6 ± 9.2 | 22.3 ± 8.5 | 0.643 |
| Hypertension | 24 (14.7%) | 2 (13.3%) | 22 (14.9%) | 1.000 |
| Fever | 82 (50.3%) | 6 (40.0%) | 76 (51.4%) | 0.402 |
| Splenomegaly | 37/157 (23.6%) | 2/14 (14.3%) | 35/143 (24.5%) | 0.598 |
| Embolic event | 14 (8.6%) | 3 (20%) | 11 (7.4%) | 0.241 |
| LVEF (%) | 61.7 ± 7.8 | 58.1 ± 13.7 | 62.1 ± 6.9 | 0.439 |
| **Valve involvement** | | | | |
| Mitral valve | 48 (29.4%) | 2 (13.3%) | 46 (31.1%) | 0.254 |
| Aortic valve | 88 (54.0%) | 11 (73.3%) | 77 (52.0%) | 0.115 |
| Mitral and Aortic valve | 16 (9.8%) | 1 (6.7%) | 15 (10.1%) | 1.000 |
| **Valve surgery** | | | | |
| Single valve | 78 (47.9%) | 8 (53.3%) | 70 (47.3%) | 0.656 |
| Multiple valve | 85 (53.3%) | 7 (46.7%) | 78 (53.9%) | |
| **Microbiology** | | | | |
| Blood culture (positive) | 46 (28.2%) | 5 (33.3%) | 41 (27.7%) | 0.872 |
| *Streptococcus* | 31 (19.0%) | 2 (13.3%) | 29 (19.6%) | 0.808 |
| **Laboratory findings** | | | | |
| WBC ($10^9$/L) | 9.4 ± 4.5 | 10.3 ± 5.2 | 9.3 ± 4.4 | 0.491 |
| HB (g/L) | 109.0 ± 23.0 | 110.5 ± 32.8 | 108.9 ± 22.0 | 0.859 |
| ALB (g/L) | 35.6 ± 5.8 | 34.2 ± 5.6 | 35.8 ± 5.8 | 0.327 |
| ALT (IU/L) | 22.0 (13.7–39.4) | 29.7 (12.1–87.3) | 21.8 (13.9–38.5) | 0.318 |
| AST (IU/L) | 23.0 (16.9–36.7) | 28.5 (17.6–77.6) | 22.8 (16.8–35.3) | 0.304 |
| CREA (umol/L) | 80.1 (67.3–105.1) | 95.6 (72.7–114.3) | 79.5 (66.5–99.3) | 0.058 |
| **Coagulation parameters** | | | | |
| PT (sec) | 13.0 ± 2.6 | 14.9 ± 4.7 | 12.8 ± 2.3 | 0.225 |
| PTA (%) | 84.2 ± 23.7 | 73.2 ± 27.7 | 85.3 ± 23.1 | 0.154 |
| PT-INR | 1.17 ± 0.23 | 1.34 ± 0.41 | 1.16 ± 0.20 | 0.193 |
| APTT (sec) | 31.9 ± 4.1 | 32.4 ± 4.2 | 31.9 ± 4.1 | 0.651 |
| TT (sec) | 14.9 ± 2.9 | 15.0 ± 2.5 | 14.9 ± 3.0 | 0.798 |
| FBG (g/L) | 3.65 ± 1.13 | 2.92 ± 0.96 | 3.73 ± 1.12 | 0.008* |
| PLT ($10^9$/L) | 182.0 (128.0–251.0) | 149.0 (84.0–212.0) | 183.0 (131.3–259.5) | 0.044* |
| D-dimer (mg/L) | 0.42 (0.18–0.99) | 0.55 (0.08–2.36) | 0.40 (0.19–0.96) | 0.974 |

**Notes:**
Data are expressed as median and interquartile range (IQR) or number and percentages (%). Abbreviations: BMI, body mass index; WBC, white blood cell; HB, haemoglobin; ALB, albumin; ALT, alanine amiotransferase; AST, aspartate aminotransferase; CREA, creatinine; PT, prothrombin time; PTA, Prothrombin activity; PT-INR, prothrombin time international normalized ratio; APTT, activated partial thromboplastin time; TT, thrombin time; FBG, fibrinogen; PLT, platelet.
* $p < 0.05$.

**Table 2 Coagulation parameters associated with all-cause mortality.**

| Parameter | Univariate analysis | Multivariate analysis | | | |
|---|---|---|---|---|---|
| | *p* value | HR | Lower limit | Upper limit | *p* value |
| BMI | 0.028* | | | | |
| Creatinine | 0.152 | | | | |
| PT-INR | 0.001* | 8.94 | 1.83 | 43.75 | 0.007* |
| APTT | 0.656 | | | | |
| TT | 0.928 | | | | |
| FBG | 0.009* | 0.55 | 0.32 | 0.94 | 0.030* |
| PLT | 0.033* | 1.00 | 0.99 | 1.01 | 0.818 |

Notes:
  *P*-value, HRs and 95% CIs were calculated by Cox proportional hazards regression models by backward conditional method; Abbreviations: PT-INR, prothrombin time international normalized ratio; APTT, activated partial thromboplastin time; TT, thrombin time; FBG, fibrinogen; PLT, platelet.
  * *p* < 0.05.

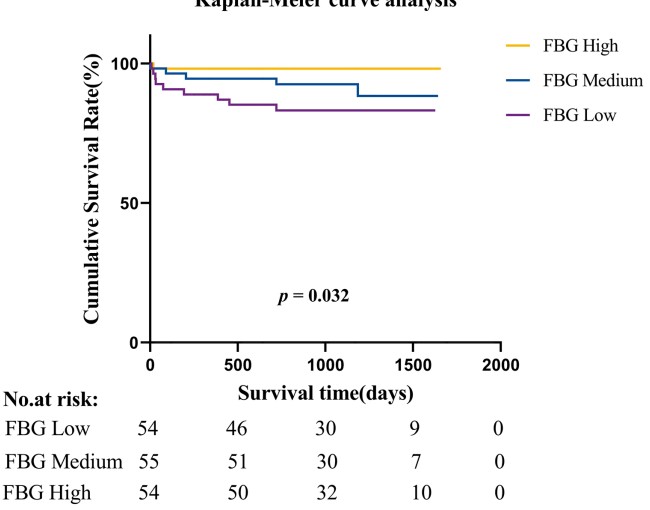

**No.at risk:**

| | | | | | |
|---|---|---|---|---|---|
| FBG Low | 54 | 46 | 30 | 9 | 0 |
| FBG Medium | 55 | 51 | 30 | 7 | 0 |
| FBG High | 54 | 50 | 32 | 10 | 0 |

**Figure 2 Kaplan-Meier survival estimates of all-cause mortality in patients with native valve infective endocarditis.**                                     

independent risk factors for all-cause mortality in the multivariate analysis. In addition, the BMI showed statistical association with all-cause mortalityin the univariate analysis (*p* = 0.028, Table 2). When the BMI was included in the multivariable Cox regression analysis, the *p* values of PT-INR and FBG remained statistically significant. Together, these data indicates that PT-INR and FBG might be potent predictors for the prognosis of NVIE patients.

Furthermore, when patients were divided into three tertile groups according to the FBG level, Kaplan–Meier curves revealed that patients with low FBG levels (<3.28 g/L) had a significantly greater mortality rate (*p* = 0.034) than those with high FBG levels (>3.99 g/L) (Fig. 2).

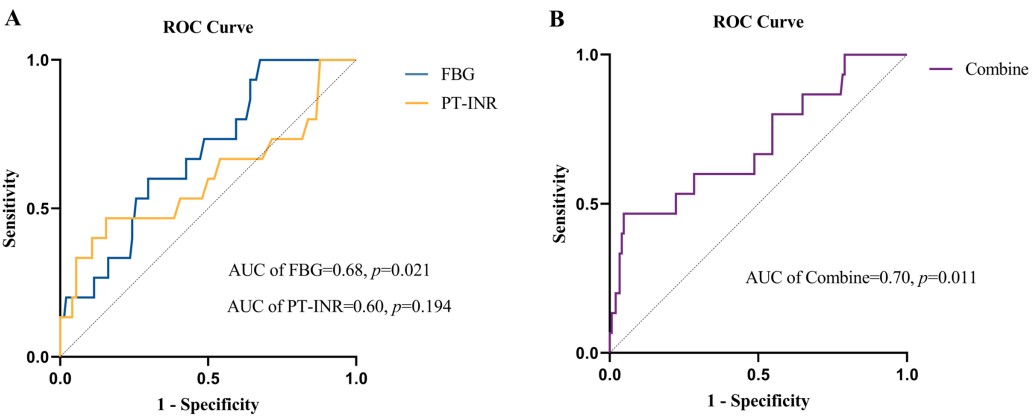

**Figure 3** (A) Receiver operating characteristic (ROC) curves of FBG level and PT-INR for predicting all-cause mortality. (B) Receiver operating characteristic (ROC) curves of FBG combined with PT-INR for predicting all-cause mortality.

**Table 3 Adjusted hazard ratios and 95% confidence interval for FBG by tertiles on all-cause mortality.**

**FBG**

|  | Q1 (*n* = 54) | Q2 (*n* = 55) | Q3 (*n* = 54) | *P* for trend |
|---|---|---|---|---|
| **Median** | 2.51 (2.13–2.75) | 3.60 (3.45–3.84) | 4.65 (4.40–5.24) | |
| **Model 1** | 1 (reference) | 0.480 (0.154–1.495) | 0.113 (0.014–0.905) | 0.018* |
| **Model 2** | 1 (reference) | 0.444 (0.139–1.416) | 0.115 (0.014–0.927) | 0.018* |
| **Model 3** | 1 (reference) | 0.666 (0.192–2.310) | 0.127 (0.016–1.032) | 0.035* |

**Notes:**
HRs and 95% CIs were calculated by Cox proportional hazards regression models by backward conditional method; *P* for trend value was gained by entering the median value of each category of FBG tertiles; Model 1 adjusted for age, gender and BMI; Model 2 adjusted for age, gender, BMI, PLT and LVEF; Model 3 adjusted for age, gender, BMI, PLT, LVEF, APTT, TT and PT-INR.
* $p < 0.05$.

## Predictive performance of coagulation parameters

The value of FBG and the PT-INR in predicting all-cause mortality was evaluated by ROC curve analysis (Fig. 3A). The AUC for FBG was 0.68 (95% CI: [0.55–0.81]; $p = 0.021$), and the cutoff value for FBG was ≤4.13 g/L, with a sensitivity of 1.00, specificity of 0.32 and Youden index of 0.32. The AUC for the PT-INR was 0.60 (95% CI: [0.42–0.78]; $p = 0.194$), and the cutoff level was ≥1.29, with a sensitivity of 0.47, a specificity of 0.84 and a Youden index of 0.31. Additionally, the AUC value of FBG combined with the PT-INR was 0.70 (95% CI: [0.55–0.85]; $p = 0.011$), which was similar to the predictive ability for all-cause mortality in NVIE patients (Fig. 3B).

Furthermore, patients were divided into three tertile groups according to the FBG level, and the prognostic value was tested *via* the linear trend test. The values of each group were substituted by the median FBG values. As shown in Table 3, the FBG tertiles were significantly related to all-cause mortality in all three adjusted models. The HR and 95% CI for Model 1 (Q3 *vs*. Q1; adjusted for age, sex and BMI) were 0.113 (95% CI: [0.014–0.905]; $p = 0.018$); those for Model 2 (Q3 *vs*. Q1; adjusted for age, sex, BMI, PLT and LVEF) were

0.115 (95% CI: [0.014–0.927]; $p$ = 0.018); and those for Model 3 (Q3 $vs$. Q1; adjusted for age, sex, BMI, PLT, LVEF, APTT, TT and PT-INR) were 0.127 (95% CI: [0.016–1.032]; $p$ = 0.035) (Table 3).

## DISCUSSION

In the present study, we evaluated the clinical significance and prognostic value of preoperative fibrinogen levels in patients with NVIE undergoing valve surgery. Our findings revealed that FBG levels in this study population are associated with all-cause mortality, and an FBG level less than 3.28 g/L was associated with a greater risk of all-cause mortality.

Some predictors of mortality associated with inflammatory biomarkers, including the erythrocyte sedimentation rate (ESR), leukocytosis, C-reactive protein (CRP) and procalcitonin (*Chu et al., 2004*; *Wallace et al., 2002*), have been shown to indicate a poor prognosis in patients with IE. Furthermore, some studies have focused on the relationships between coagulation parameters and IE. In a study including 337 patients with IE, *Zampino et al. (2021)* showed that elevated D-dimer levels and prolonged PT-INR were associated with a greater risk for in-hospital mortality in IE patients, whereas a longer APTT was associated with 1-year mortality. We observed similar results in our cohort: a prolonged PT-INR was associated with a higher risk for all-cause mortality. However, the PT-INR showed poor predictive performance because of the absence of statistical significance in the ROC curve. The inconsistent results between the Cox model and the ROC curve may be due to the low incidence of endpoint events and the small sample size of the cohort. *Turak et al. (2014)* conducted a prospective cohort study including 157 patients with IE and reported that D-dimer ≥4.2 mg/L was significantly associated with increased in-hospital mortality. Another retrospective observational study included 414 patients revealed a higher BMI was independent predictor of long-term mortality in patients with left-sided infective endocarditis (*Durante-Mangoni et al., 2021*). This finding supported the observation that BMI was associated with all-cause mortality in our cohort. Together with BMI, FBG and PT-INR exerted good predicative performance on the all-cause mortality in patients with NVIE. Nevertheless, the lack of external validation decreased the level of evidence rank.

The conversion of fibrinogen into insoluble fibrin contributes to the formation of blood clots and prevents further bacterial spread by sealing infected tissue with a fibrin network (*Durante-Mangoni, Molaro & Iossa, 2014*; *Mosesson, 2005*). This strong interaction between the coagulation system and immune system is referred to as immunothrombosis, which is a highly balanced system with an adequate response to infective stimuli. However, in the case of IE, this balance is often disrupted (*Liesenborghs et al., 2020*). The normal range of the plasma fibrinogen level is approximately 2–5 g/L. Notably, fibrinogen is an acute phase protein that can be upregulated by inflammatory reactions in IE (*Werdan et al., 2014*). Some patients in our cohort presented with a decrease in fibrinogen levels and showed a tendency toward consumption coagulopathy. The risk of major bleeding is markedly increased when plasma fibrinogen levels are reduced below 1.5 to 2 g/L

(*Rossaint et al., 2016*). The supplementation of fibrinogen concentrate may reduce bleeding rates and improve treatment safety in cardiac surgery patients (*Grottke et al., 2020*). These findings may explain the results of our study, as the bottom tertile of the FBG level was associated with a greater risk than the top tertile was.

Prior studies have shown that early surgery is associated with a 40% to 60% reduction in all-cause death compared with conventional therapy (antibiotic treatment or late surgery) (*Wang, Gaca & Chu, 2018*; *Habib et al., 2019*). A survey conducted in France revealed that over 70% of NVIE patients had surgery indications according to the guidelines, 46% of whom underwent surgery during the acute phase (*Delahaye, 2011*). This may raise concerns about the possible insufficient use of cardiac surgery in NVIE. Some specific risk scores have been developed to assess the risk of surgery. The predictive performance of these scores is questioned in the particular case of early surgery for NVIE (*Gaca et al., 2011*; *Martínez-Sellés et al., 2014*). More promising indicators and scores need to be developed and validated to predict the risk and prognosis after surgery. To the best of our knowledge, no prior similar studies focused on FBG and NVIE exist. Therefore, we believe that the FBG level should be evaluated immediately at admission because of the effective predictive performance shown above. When the plasma concentration of FBG is below the cutoff value that we have provided, the timing of surgery should be reconsidered because of the elevated risk of all-cause mortality. Our results may help surgeons assess the risk and prognosis of early valve surgery in patients with NVIE.

Some limitations of the study should be taken into account. First, this was a small cohort due to the low incidence of NVIE and the retrospective, observational and single-center nature of the study. Second, we focused mainly on coagulation parameters, which may limit the identification of other valuable parameters. Third, the decision to perform surgery was made on the basis of the clinical judgment of the surgical team, which may cause bias in the selection of therapeutic strategies. Furthermore,the interaction effect between PT-INR and FBG was not be fully excluded due to the limited sample size. Additionally, low FBG levels was related to disseminated intravascular coagulation (DIC). Although none of the patients in this cohort were diagnosed with DIC, coagulation dysfunction might occur in the microcirculation system and DIC might be underestimated. Finally, the cause-and-effect relationship between lower FBG and poor NVIE prognosis could not be concluded, which limited the extension of this observation to other NVIE scenarios. Further research remains needed to obtain a better understanding of the relationship between FBG and NVIE.

## CONCLUSION

Our study results suggest that the preoperative FBG level is a good prognostic factor for all-cause mortality in NVIE patients undergoing surgery and that an FBG level less than 3.28 g/L indicates a greater risk of all-cause mortality. Our results may help evaluate the prognosis of NVIE patients underwent surgery and identify those who might benefit from early operation.

### Funding

This study were supported by the Chongqing Medical Scientific Research Project (Joint Project of Chongqing Health Commission and Science and Technology Bureau) (No. 2022MSXM115) and the Chongqing National Science Foundation General Project (CSTB2022NSCQ-MSX1249). The funders had no role in study design, data collection and analysis, decision to publish, or preparation of the manuscript.

### Grant Disclosures

The following grant information was disclosed by the authors:
Chongqing Medical Scientific Research Project (Joint Project of Chongqing Health Commission and Science and Technology Bureau): 2022MSXM115.
Chongqing National Science Foundation General Project: CSTB2022NSCQ-MSX1249.

### Competing Interests

The authors declare that they have no competing interests.

### Author Contributions

- Jia Li conceived and designed the experiments, performed the experiments, analyzed the data, prepared figures and/or tables, authored or reviewed drafts of the article, and approved the final draft.
- Junyong Zhao conceived and designed the experiments, performed the experiments, analyzed the data, prepared figures and/or tables, authored or reviewed drafts of the article, and approved the final draft.
- Ning Sun performed the experiments, prepared figures and/or tables, and approved the final draft.
- Lijiao Zhang performed the experiments, prepared figures and/or tables, and approved the final draft.
- Qing Su performed the experiments, prepared figures and/or tables, and approved the final draft.
- Wei Xu performed the experiments, prepared figures and/or tables, and approved the final draft.
- Xiaolin Luo analyzed the data, prepared figures and/or tables, and approved the final draft.
- Zhichun Gao analyzed the data, prepared figures and/or tables, and approved the final draft.
- Keting Zhu analyzed the data, prepared figures and/or tables, and approved the final draft.
- Renjie Zhou conceived and designed the experiments, authored or reviewed drafts of the article, and approved the final draft.
- Zhexue Qin conceived and designed the experiments, authored or reviewed drafts of the article, and approved the final draft.

## Human Ethics

The following information was supplied relating to ethical approvals (*i.e.*, approving body and any reference numbers):

The study was approved by the Medical Ethics Committee of Second Affiliated Hospital of Army Medical University (No. 2023-Research 109-01).

## Data Availability

The raw measurements are available in the Supplemental File.

## Supplemental Information

Supplemental information for this article can be found online at http://dx.doi.org/10.7717/peerj.18182#supplemental-information.

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
