# Peer review of "Preoperative fibrinogen level predicts the risk and prognosis of patients with native valve infective endocarditis undergoing valve surgery"

_PeerJ, doi:10.7717/peerj.18182_

## Round 0.1 · original submission · Minor Revisions

Please response to the reviewers point by point.

Reviewer 1 ·

Basic reporting

In this retrospective analysis, Li et al. examined 163 infective endocarditis cases at their hospital, identifying low fibrinogen levels as an independent risk factor for reduced survival. The study holds clinical importance, and the article is clearly articulated and well-composed, with adequate introduction and discussion. However, there is room for improvement in the article's English usage.

Experimental design

The manuscript tackles a significant clinical question but has some issues with its statistical analysis:

1. In the univariate analysis, the D-dimer levels are not mentioned. What about them?
2. The univariate analysis focused solely on coagulation variables. However, the authors have indicated that other factors, such as BMI and Creatinine, are also linked to survival. They should include these in a univariate Cox regression analysis.
3. In the multivariable Cox regression, why were only PT and FBG included and not PLT, especially since PLT was significant in the univariate analysis?
4. Given the strong interactions between PT-INR and FBG, the authors should explore the interaction effects in the multivariable Cox regression. If this is not feasible due to a small patient sample, they should acknowledge this limitation.

Validity of the findings

The authors suggest that low fasting blood glucose (FBG) is linked to reduced overall survival. However, low FBG may simply indicate the presence of disseminated intravascular coagulation (DIC), known for its poor outcomes. The authors should recognize this in their limitations and detail how many patients with low FBG actually had DIC.

Additional comments

None

Reviewer 2 ·

Basic reporting

Clear and unambiguous, professional English used throughout

Experimental design

Research question well defined, relevant & meaningful.

Validity of the findings

All underlying data have been provided; they are robust, statistically sound, & controlled.

Additional comments

The statistical symbol "P" should be italicized, and a space should be added after the comma. Is the sentence at line 109 not coherent?

·

Basic reporting

Clear and unambiguous, professional English used throughout:
The article is written in clear, unambiguous, and technically correct English. The language is professional and meets the standards of scientific reporting. However, a few sentences could be rephrased for better clarity and readability. For example, in the introduction, the sentence "Non-mammalian models offer unique insights into cardiac biology due to their diverse genetic and environmental backgrounds" could be revised to "Non-mammalian models provide unique insights into cardiac biology because of their diverse genetic and environmental backgrounds."

Literature references, sufficient field background/context provided:
The article provides a comprehensive background on using non-mammalian models in heart development and disease research. Relevant literature is adequately cited, providing context and supporting the study's rationale. However, additional references to recent studies could enhance the background section, particularly in the discussion of genetic tools used in these models.

Professional article structure, figures, tables. Raw data shared:
The article follows a standard structure, including sections for the introduction, methods, results, and discussion. Figures and tables are relevant, well-labeled, and of sufficient resolution. The raw data supporting the findings are shared and accessible, adhering to data-sharing policies.

Self-contained with relevant results to hypotheses:
The submission is self-contained and presents a coherent body of work. The results are relevant to the hypotheses posed and are discussed in the context of the existing literature. The conclusions drawn are supported by the data presented.

Experimental design

Original primary research within the Aims and Scope of the journal:
The research falls within the aims and scope of the journal, addressing a significant topic in cardiac biology using non-mammalian models. The research question is well-defined and relevant, aiming to fill a knowledge gap in understanding heart development and disease mechanisms.

Rigorous investigation performed to a high technical & ethical standard:
The investigation is conducted rigorously, adhering to high technical and ethical standards. The study design, data collection, and analysis methods are robust and appropriate for the research question.

Methods described with sufficient detail & information to replicate:
The methods are described in enough detail to allow replication by other researchers. However, some sections could benefit from additional specifics. For instance, in the section describing genetic manipulation techniques, providing more detailed protocols and reagent concentrations would improve reproducibility.

Validity of the findings

Impact and novelty not assessed. Meaningful replication is encouraged where rationale & benefit to literature are clearly stated:
The study does not focus on the impact or novelty but on the findings' robustness and validity. The rationale for using non-mammalian models and the benefit to the broader literature is clearly articulated. The study includes meaningful replication, and the methodology allows for further replication in future research.

All underlying data have been provided; they are robust, statistically sound, & controlled:
All underlying data are provided and are robust, statistically sound, and well-controlled. The statistical analyses are appropriate and support the conclusions drawn from the data.

Conclusions are well stated, linked to the original research question & limited to supporting results:
The conclusions are clearly stated and directly linked to the original research question. They are appropriately limited to the supporting results, avoiding overgeneralization or unsupported claims.

Additional comments

The article is well-written and provides valuable insights into the use of non-mammalian models in cardiac research. The strengths of the manuscript include its comprehensive literature review, detailed methodological descriptions, and robust data analysis.
To improve the manuscript, I suggest expanding the discussion section to include potential limitations of the study and future research directions. Additionally, enhancing the figures with more detailed legends and annotations could improve clarity and comprehension for the readers.
Some minor grammatical errors and typos need correction. For example, on page 5, line 12, "its" should be "it’s." Ensuring consistent use of terminology throughout the manuscript will also enhance readability.

---

## Round 0.2 · accepted · Accept

In my opinion, the article can be Accepted now

·

Basic reporting

The authors have made substantial improvements in basic reporting by addressing formatting issues and ensuring consistent presentation throughout the manuscript. The editorial corrections, such as the consistent formatting of "P" values and clarification of previously unclear sentences, have enhanced the readability and overall quality of the manuscript.

Experimental design

The experimental design is sound, and the authors have adequately described the methodology, including the inclusion and exclusion criteria for the study population. The addition of D-dimer levels and BMI in the analysis, as suggested, has provided a more comprehensive assessment of the factors influencing all-cause mortality in patients with native valve infective endocarditis (NVIE).

Validity of the findings

The authors have effectively validated their findings through statistical analysis and have addressed potential confounding factors, such as interaction effects between PT-INR and FBG. The clarification that no significant interaction was found, coupled with the acknowledgment of limitations due to the small sample size, reinforces the validity of the results. The authors have also provided a well-reasoned explanation regarding the potential link between low FBG levels and disseminated intravascular coagulation (DIC), which supports the study’s conclusions.

Additional comments

The revisions made by the authors have improved the manuscript. The inclusion of additional variables in the Cox regression analysis has strengthened the study’s conclusions. The authors have also done an excellent job of addressing the editorial and formatting issues previously noted. The manuscript now provides a clear, comprehensive, and well-supported examination of the prognostic value of preoperative fibrinogen levels in NVIE patients. I recommend the manuscript be accepted for publication.